# Fabrication of Rapidly Soluble Zn^2+^-Releasing Phosphate-Based Glass and Its Incorporation into Dental Resin

**DOI:** 10.3390/molecules29215098

**Published:** 2024-10-29

**Authors:** Fan Deng, Haruaki Kitagawa, Tomoki Kohno, Tingyi Wu, Naoya Funayama, Pasiree Thongthai, Hefei Li, Gabriela L. Abe, Ranna Kitagawa, Jun-Ichi Sasaki, Satoshi Imazato

**Affiliations:** 1Department of Dental Biomaterials, Osaka University Graduate School of Dentistry, 1-8 Yamadaoka, Suita 565-0871, Japanu829939b@ecs.osaka-u.ac.jp (T.W.); pasiree.t@chula.ac.th (P.T.); kitagawa.ranna.dent@osaka-u.ac.jp (R.K.); sasaki.junichi.dent@osaka-u.ac.jp (J.-I.S.); imazato.satoshi.dent@osaka-u.ac.jp (S.I.); 2Joint Research Laboratory of Advanced Functional Materials Science, Osaka University Graduate School of Dentistry, 1-8, Yamadaoka, Suita 565-0871, Japanfunayama.naoya.dent@osaka-u.ac.jp (N.F.); lihefei115@gmail.com (H.L.); abe.gabriela.dent@osaka-u.ac.jp (G.L.A.); 3Department of Operative Dentistry, Faculty of Dentistry, Chulalongkorn University, Pathum Wan, Bangkok 10330, Thailand

**Keywords:** dental materials, ion-releasing glass, zinc, deep caries, liner

## Abstract

Phosphate-based glasses are known for their excellent biocompatibility and adjustable degradation rates. In this study, we fabricated a rapidly soluble zinc-ion-releasing phosphate-based glass (RG) specifically designed for use in dental cavity liners. The aim of this study was to evaluate the ion-releasing properties and antibacterial effects of RG. Additionally, a dental resin incorporating RG was fabricated to serve as a cavity liner, and its effectiveness was investigated in vitro. The RG formulation exhibited high solubility and released high concentrations of Zn^2+^ at various pH values. To assess the antibacterial properties of RG, six bacterial species detected in deep carious regions were incubated in the presence of RG. In vitro antibacterial testing against six bacterial species revealed that RG exhibited strong bactericidal effects against these prevalent bacteria. Furthermore, using a dentin model infected with *Lactobacillus casei* or *Streptococcus mutans*, the experimental resin containing RG demonstrated an effective bactericidal effect in the dentinal tubules, highlighting its potential as a promising material for cavity liners or pulp-capping applications.

## 1. Introduction

Despite advancements in preventive and therapeutic strategies, dental caries remain a significant challenge for both oral health professionals and patients [1]. Delayed or inadequate treatment of early caries can result in its progression beyond the enamel, reaching the dentin and pulp, thereby increasing the likelihood of deep caries [2]. For the treatment of deep caries, selective (incomplete) caries removal is widely recommended, a technique that involves partially removing carious dentin while leaving some affected dentin intact to avoid pulp exposure [3,4]. In addition to the less invasive technique, the application of cavity liners or indirect pulp-capping materials is often proposed to further protect the dentin and promote healing [5,6]. Calcium hydroxide (Ca(OH)_2_) is one of the most commonly employed liners and pulp-capping agents, known for its antibacterial properties, which are believed to result from the creation of a localized alkaline pH upon dissolution [7,8]. However, the efficacy of Ca(OH)_2_-based materials has been questioned in several studies, as their antibacterial effects may not be sufficiently strong to achieve bactericidal action on dentin [9,10]. Given the potential risk of bacteria remaining after selective caries removal, achieving a sterile cavity remains a critical goal.

Recently, several attempts have been made to produce ion-releasing glasses with antibacterial properties by incorporating zinc (Zn) [11,12,13]. Zn^2+^ has multiple inhibitory effects on the physiology of intact bacterial cells, including glycolysis, ATPase activity, and sugar phosphotransferases [14,15,16]. It enhances proton permeability in bacterial cell membranes, reduces acid tolerance, and disrupts cell metabolism [17,18]. Previously, we developed an acidity-responsive silicate-based glass containing Zn (AG) that exhibited on-demand Zn^2+^ release properties [19]. However, under the conditions required for cavity disinfectants, the Zn^2+^ release from AG is insufficient to achieve the high concentration of Zn^2+^ needed to effectively kill bacteria in dentinal tubules, due to the relatively low solubility of silicate-based glass.

Phosphate-based glass is a promising new-generation biomaterial due to its excellent biocompatibility and versatility, allowing for the design of materials optimized for various medical applications [20]. In engineering applications, phosphate-based glass rapidly dissolves under wet conditions, irrespective of whether the environment is acidic or alkaline [20]. This study presents a rapidly soluble Zn^2+^-releasing phosphate-based glass (RG), hypothesized to quickly release a high concentration of Zn^2+^ and exhibit cavity-disinfecting effects. In this study, the ion-releasing properties and antibacterial effects of RG were examined. Additionally, a dental resin incorporating RG as a cavity liner was fabricated, and its antibacterial efficacy was evaluated using infected dentin models.

## 2. Results

### 2.1. Characteristics of Rapidly Soluble Zn^2+^-Releasing Glass (RG)

Field-emission scanning electron microscope (FE-SEM) images of the RG particles revealed that the glass had an irregular shape (Figure 1a). Energy dispersive spectrometry (EDS) mapping images confirmed that P, Na, Ca, and Zn were homogeneously dispersed within the particles (Figure 1b). The size distribution of RG particles, shown in Figure 2, indicated a median diameter of approximately 11.95 µm. The X-ray diffraction (XRD) pattern confirmed that the sample structure was amorphous (Figure 3). X-ray fluorescence (XRF) analysis detected phosphorus pentoxide (P_2_O_5_), sodium oxide (Na_2_O), calcium oxide (CaO), and zinc oxide (ZnO) in RG at 41.2, 40.4, 4.5, and 11.9 mol%, respectively. No remarkable differences in composition were observed between the preparation and XRF analysis results.

### 2.2. Solubility of RG

Figure 4 shows the solubility of RG particles in a pH-adjusted buffer solution (pH 7.0, 6.0, or 5.0). The dissolution rates at pH 7.0, 6.0, and 5.0 were 98.7 ± 0.31%, 96.4 ± 0.63%, and 93.8 ± 0.92%, respectively. Across all three pH conditions, the solubility of RG after 24 h was greater than 90%.

### 2.3. Ion-Releasing Property of RG

The concentration of Zn^2+^ released from RG into the pH-adjusted BHI broth (pH 7.0, 6.0, or 5.0) is illustrated in Figure 5. The Zn^2+^ concentrations released from RG at pH 7.0, 6.0, and 5.0 were 8172.5 ± 299.8 ppm, 6851.6 ± 98.2 ppm, and 5978.5 ± 63.7 ppm, respectively. Under all three pH conditions, the concentrations of released Zn^2+^ exceeded 5900 ppm.

### 2.4. Minimum Inhibitory Concentrations (MICs) and Minimum Bactericidal Concentrations (MBCs) of Zn^2+^ Against Oral Bacteria

Table 1 lists the MICs and MBCs of Zn^2+^ against *L. casei*, *A. naeslundii*, *E. faecalis*, *P. micra*, *F. nucleatum* and *S. mutans*. For the six species, the MIC values ranged from 62.5 to 1000 ppm, while the MBC values ranged from 125 to 4000 ppm.

### 2.5. Antibacterial Activity of RG Against Oral Bacteria

Figure 6 shows the number of viable bacteria after incubation in the presence of RG particles. A significant decrease in the number of surviving cells was observed for all bacterial species tested. Although RG particles demonstrated only a 3–4 log reduction in the bactericidal effect against *E. faecalis*, they showed a 100% killing effect against all other bacteria.

### 2.6. Inhibition of Bacteria in Dentinal Tubules by Dental Resin Incorporating RG

Confocal laser scanning microscopy (CLSM) analysis confirmed that the depth of *L. casei* penetration in the dentinal tubules reached approximately 148 ± 80 µm (Figure 7a). Some viable bacteria remained in the dentinal tubules after RG treatment when the dentin specimens were dried (Figure 7b). In contrast, RG-containing dental resin completely eradicated *L. casei* in the wet state from the dentinal tubules (Figure 7c), whereas the control resin without RG did not exhibit antimicrobial effects against *L. casei* in the dentinal tubules (Figure 7d).

The depth of *S. mutans* penetration in the dentinal tubules was approximately 212 ± 71 µm (Figure 8a). Following RG treatment, some viable *S. mutans* remained when the specimens were dried (Figure 8b). However, the RG-containing dental resin fully eradicated *S. mutans* under wet conditions (Figure 8c), while the control resin without RG did not show antimicrobial effects (Figure 8d).

## 3. Discussion

The P-O-P bonds in phosphate-based glasses, which form the glass network structure, are highly susceptible to proton attack and are easily hydrolyzed in aqueous media [21]. The addition of cations such as Zn^2+^ enhances glass formation by strengthening the network and suppressing crystallization, thus expanding the glass-forming region of polyphosphate glasses [22]. In addition, ZnO contributes to the chemical stability of phosphate glass by forming [ZnO_4_] tetrahedra or P-O-Zn linkages within the glass network structure [23]. Previous studies have noted that pyrophosphates (P_2_O_7_^4−^) are present in phosphate-based glasses when the P_2_O_5_ content is less than 50 mol% [24]. Pyare et al. [25] observed increased solubility of phosphate glasses with 50 mol% P_2_O_5_ as the pH increased, attributing this to hydroxyl ions (OH^−^) breaking P-O-P bonds through the hydrolysis reaction: P_2_O_7_^4−^ + 2OH^−^ → 2PO_4_^3−^ + H_2_O. Even in the presence of network modifiers such as ZnO, the Zn-O-P linkages are susceptible to OH^−^ attack through the following reaction: Zn^2+^⋯P_2_O_7_^4−^⋯Zn^2+^ + 2OH^−^ → 2Zn^2+^ + 2PO_4_^3−^ + H_2_O. In this study, the mol fraction of P_2_O_5_ in RG was 41.2 mol%, indicating a glass structure containing P_2_O_7_^4−^ units. This structure contributed to the reduced solubility and Zn^2+^ release with decreasing pH. Nevertheless, RG exhibited solubility over 90% after 24 h and Zn^2+^ concentrations exceeding 5900 ppm across all tested pH conditions, demonstrating its ability to release high concentrations of Zn^2+^ rapidly.

According to numerous reports on the microbiome associated with deep caries, *Lactobacillus* spp., *Actinomyces* spp., *Fusobacterium* spp., and *Streptococcus* spp. are prevalent in deep carious lesions [26,27,28,29]. Additionally, *P. micra* has been detected in carious dentin [30] and irreversible pulpitis [31,32], and *E. faecalis* has been isolated from deep carious lesions and infected root canals [33,34]. The MIC values of Zn^2+^ against *L. casei*, *A. naeslundii*, *F. nucleatum*, *P. micra* and *E. faecalis* were 500, 62.5, 125, 125, and 1000 ppm, respectively, and the MBC values were 2000, 125, 500, 250, and 4000 ppm, respectively. RG particles released Zn^2+^ concentrations exceeding 5900 ppm under pH conditions of 5.0, 6.0, and 7.0, which was significantly greater than the MIC and MBC values for each tested bacterium. Therefore, the amount of Zn^2+^ released from the RG particles was sufficient to exert a bactericidal effect on the bacterial species. However, RG particles demonstrated only a 3–4 log reduction in bactericidal effect against *E. faecalis*, while achieving a 100% killing effect against all other bacteria. Unlike the other bacterial species tested in this study, *E. faecalis*, a natural member of the intestinal flora, can survive in a wide range of pH, heat, and high metal concentrations [35]. Zincophore biosynthetic gene clusters have been identified in various bacteria, including Actinobacteria, Clostridia, Fusobacteria, and Bacilli, suggesting potential targets for antimicrobial therapies [36]. Likewise, Zn-responsive proteins have also been identified in *E. faecalis* genes, which mediate bacterial defense against Zn overload [37]. This suggests that *E. faecalis* is less affected by Zn^2+^ released from RG particles compared to other bacteria. Due to its ability to survive and persist as a pathogen in dentinal tubules, extensive research has been conducted to find effective methods to eradicate *E. faecalis*, and the antimicrobial efficacy of Zn-containing sealers against *E. faecalis* has been demonstrated [38].

The level of bacterial infection following dental caries removal varies clinically. Hirose et al. [39] created dentin models with varying infection levels by culturing dentin specimens in *S. mutans* suspension for 6 or 12 h. These models exhibited bacterial penetration into the dentinal tubules up to approximately 50 µm from the surface. Kitagawa et al. [40] prepared infected dentin specimens by immersing root dentin in *E. faecalis* suspension for 2 d, resulting in penetration of approximately 200 µm into the dentinal tubules. In this study, a dentin model infected with *L. casei*, a prevalent species in deep carious lesions, was established by culturing dentin specimens in *L. casei* suspension. Preliminary experiments showed no clear bacterial penetration into the dentinal tubules after 2 d of incubation. Consequently, dentin specimens were immersed in *L. casei* suspension for 7 d, resulting in *L. casei* penetrating approximately 150 µm into the dentinal tubules. Similarly, dentin specimens immersed in *S. mutans* suspension for 7 d exhibited bacterial penetration up to about 200 µm. There was no significant difference in penetration depth between *L. casei* and *S. mutans*. Factors influencing bacterial penetration include the status of dentinal tubules (size and orientation), microorganism size, and bacterial motility [41,42]. *L. casei* is nonmotile, rod-shaped (0.7–1.1 µm × 2.0–4.0 µm), and often exists singly or in short chains [43]. Its rod shape facilitates entry into the dentinal tubules, though its motility is relatively limited compared to other bacteria. In contrast, *S. mutans* are spherical (0.5–0.75 µm in diameter) and often found in pairs or chains [44]. Both bacteria are sufficiently small to enter dentinal tubules (3.47 ± 0.73 µm for permanent teeth) [45], and their penetration is likely driven by diffusion and passive processes rather than active movement [46]. Acid production by both bacteria can demineralize dentin, potentially enlarging the tubules and aiding in deeper penetration. The similarity in penetration depth suggests that factors other than morphology, such as acid production and the intrinsic properties of dentin, play a more critical role in bacterial infiltration of dentinal tubules. Therefore, using *S. mutans* as a reference for *L. casei* is crucial, as it validates the infection model for assessing the efficacy of dental resins incorporating RG in inhibiting bacterial growth in dentinal tubules.

The antibacterial effects of the experimental resins were evaluated using established infected dentin models. Under wet conditions, the bacteria in the dentin were effectively killed, whereas the dry state showed less efficacy. This difference is attributed to the greater Zn^2+^ release under wet conditions, due to the dissolution of the glass in the experimental resin. The RG-incorporating resin was designed as a cavity liner or pulp-capping material, which typically operates in moist environments. Under such conditions, Zn^2+^ is rapidly released and penetrates deeply into the dentinal tubules, resulting in swift bacterial eradication.

However, the infected dentin model used in this study did not fully replicate the complex environment of the dentinal cavity, which harbors multiple bacterial species. In vivo studies are essential for assessing the clinical effectiveness of RG-incorporating resins and for optimizing their composition. Additionally, it is important to investigate whether the rapid solubility of the glass might lead to potential drawbacks, such as weakening of the resin structure or adverse reactions in the surrounding tissues over time.

## 4. Materials and Methods

### 4.1. Fabrication of RG

Rapidly soluble phosphate-based glass containing Zn (RG) was fabricated using the melt-quenching method [47]. Briefly, phosphoric acid (H_3_PO_4_), sodium carbonate (Na_2_CO_3_), calcium carbonate (CaCO_3_), and zinc oxide (ZnO), purchased from FUJIFILM Wako Pure Chemical Corporation (Osaka, Japan), were mixed and melted in the furnace at 1100 °C for 1 h. The melted glass was quenched at 25 °C and pressed using an iron plate to obtain cullets. The glass cullets were ground into particles with ethyl alcohol using a rotating ball mill (Pot Mill Rotary Stand, Nitto Kagaku, Nagoya, Japan). The ground particles were passed through a stainless-steel sieve (250 µm mesh testing sieve, Nonaka Rikaki) to exclude insufficiently ground particles. The mixing ratios of the components were 42.0 P_2_O_5_, 40.0 Na_2_O, 4.0 CaO, and 12.0 ZnO in the mol fraction. The glass particles were sterilized using ethylene oxide.

### 4.2. Characterization of RG

High-resolution imaging of the RG surface was performed using FE-SEM (JSM-F100, JEOL, Tokyo, Japan), with elemental mapping performed via EDS (JSM-F100, JEOL). The glass structure was confirmed using XRD (PIXcel1D, Malvern Pan-Analytical Ltd., Worcestershire, UK). The size distribution of the glass particles was determined with a particle size analyzer (Partica LA-960V2, Horiba, Kyoto, Japan). The elemental composition of the glass was analyzed by XRF (ZSX Primus IV𝒾, Rigaku, Tokyo, Japan).

### 4.3. Evaluation of Solubility of RG

The solubility of RG particles in buffer solutions with varying pH values was assessed. Buffer solutions were prepared with 4-(2-hydroxyethyl)-1-piperazineethanesulfonic acid (HEPES; FUJIFILM Wako Pure Chemical) at pH 7.0, and with acetate buffer (Kanto Chemical, Tokyo, Japan) at pH 6.0 and 5.0. Fifty milligrams of RG particles were immersed in 10 mL of each buffer solution. After 24 h of gyratory shaking at 37 °C and 100 rpm, the suspensions were filtered through a 55 mm glass fiber filter with a pore size of 0.5 μm (ADVANTEC, Tokyo, Japan), which had been pre-weighed. The filter was then dried at 110 °C for 6 h. The total weight of the glass fiber filter with the undissolved glass particles was measured, and the solubility of RG was calculated. The experiment was repeated five times.

### 4.4. Evaluation of Ion-Releasing Property of RG

The concentration of Zn^2+^ released from RG particles into pH-adjusted BHI broth was assessed. The pH of the BHI broth was adjusted to 7.0, 6.0, or 5.0 by adding 1 mol/L hydrochloric acid (Kanto Chemical). Twenty milligrams of RG particles were placed in each well of a 96-well microplate, and 200 μL of pH-adjusted BHI broth was added to the wells containing the glass particles. After 24 h of gyratory shaking at 37 °C and 100 rpm, the suspensions were filtered through a 0.22 μm syringe filter (Merck Millipore Ltd., Carrigtwohill, Ireland) and diluted with 9.8 mL of distilled water. The diluted samples were analyzed for Zn^2+^ concentration using inductively coupled plasma–optical emission spectrometry (ICP-OES; iCAP7200 ICP-OES Duo, Thermo Fisher Scientific, Cambridge, UK). The experiment was repeated five times.

### 4.5. Bacteria Used

*Lactobacillus casei* ATCC4646, *Actinomyces naeslundii* ATCC19246, *Enterococcus faecalis* SS497, *Fusobacterium nucleatum* 1436, and *Parvimonas micra* GIFU7745 were selected based on their presence in deep caries (Table 2). *Streptococcus mutans* NCTC10449 was used as the reference strain. Prior to the experiments, each bacterium was anaerobically cultured from a stock culture at 37 °C using the media specified in Table 2. *E. faecalis* was cultured for 12 h, *A. naeslundii* for 48 h, and *L. casei*, *F. nucleatum*, *P. micra*, and *S. mutans* for 24 h.

### 4.6. Measurements of MICs and MBCs of Zn^2+^ Against Oral Bacteria

The MIC and MBC values of Zn^2+^ against *L. casei*, *A. naeslundii*, *E. faecalis*, *F. nucleatum*, and *P. micra* were determined using the broth dilution method [48]. Zn standard solutions were obtained by dissolving zinc nitrate hexahydrate (Zn(NO_3_)_2_·6H_2_O; Kanto Chemical) into distilled water. Fifty microliters of these solutions, with concentrations ranging from 32.5 to 32,000 ppm obtained through serial two-fold dilutions, were poured into the wells of a 96-well microplate. Fifty microliters of each bacterial suspension, adjusted to approximately 2.0 × 10^6^ CFU/mL in double-concentration broth, was poured into each well, resulting in Zn^2+^ concentrations ranging from 16.25 to 16,000 ppm and bacterial suspensions of 1 × 10^6^ CFU/mL. After anaerobic incubation at 37 °C for 24 h, turbidity was assessed visually to determine the MIC, defined as the lowest concentration that prevents visible bacterial growth. Twenty microliters of samples showing no visible turbidity were inoculated onto agar plates. After anaerobic incubation at 37 °C for 48 h, the MBC was determined as the lowest concentration that resulted in no bacterial colonies on the agar plates. The experiment was repeated five times.

### 4.7. Evaluation of Antibacterial Activity of RG Against Oral Bacteria

The antibacterial activity of RG against *L. casei*, *A. naeslundii*, *E. faecalis*, *F. nucleatum*, *P. micra,* and *S. mutans* was also evaluated. Each bacterial suspension was adjusted to approximately 1 × 10^6^ CFU/mL in the respective broths listed in Table 2. Twenty milligrams of RG was placed in a well of a 96-well microplate, and 200 μL of each bacterial suspension was poured into the wells containing RG particles. After anaerobic incubation at 37 °C for 24 h, 100 μL of each suspension was collected and diluted with 9.9 mL of the corresponding broth. The suspensions were further serially diluted with the respective broths, and 100 μL of each dilution was inoculated onto agar plates specified in Table 2. The number of colonies on the agar plates was counted after anaerobic incubation at 37 °C for 48 h. Bacterial suspensions incubated without RG served as controls. The experiment was repeated five times.

### 4.8. Fabrication of Dental Resin Incorporating RG

The resin was prepared by mixing RG with triethylene glycol dimethacrylate (TEGDMA), 2-hydroxyethyl methacrylate (HEMA), camphoroquinone (CQ), ethyl p-dimethylaminobenzoate (EPA), and 2,6-di-tert-butyl-p-benzoyl (BHT) (Table 3). RG particles and uncured resin liquid were mixed at a ratio of 1:9 (*w*/*w*), providing 10 wt% RG in the experimental resin. A resin without RG was used as a control.

### 4.9. Evaluation of Inhibition of Bacteria in Dentinal Tubules by Dental Resin Incorporating RG

To evaluate the inhibitory effect of dental resins incorporating RG on bacteria in dentinal tubules, infected dentin models were prepared using the method described by Kitagawa et al. [23]. Extracted human sound molars were obtained from patients at Osaka University Dental Hospital under a protocol approved by the Ethics Review Committee of Osaka University Graduate School of Dentistry and Osaka University Dental Hospital (approval number: R1-E52). The teeth were sectioned using a low-speed diamond saw (Isomet 2000, Buehler, Lake Bluff, IL, USA) with water cooling, and rectangular parallelepiped specimens were cut from the coronal dentin approximately 1 mm above the coronal pulp. The specimens were then adjusted to dimensions of 4 mm in length, 2 mm in width, and 2 mm in height using a grinder (EcoMet 3000, Buehler).

To remove the smear layer and open the dentinal tubules, dentin specimens were sequentially immersed in 5.25% sodium hypochlorite (Neo Dental Chemical Products Co., Tokyo, Japan) and 18% ethylenediaminetetraacetic acid (EDTA; Ultradent Products, South Jordan, UT, USA) for 10 min, with ultrasonication at 37 kHz. The specimens were then autoclaved at 120 °C for 20 min and anaerobically incubated in BHI broth at 37 °C for 48 h to confirm sterilization. The bottom and side surfaces of the specimens were covered with two layers of nail varnish.

Suspensions of *L. casei* and *S. mutans* were adjusted to approximately 1 × 10^6^ CFU/mL. The specimens were anaerobically incubated in 5 mL of each bacterial suspension at 37 °C, with the culture medium being exchanged every 24 h. After 7 d, the inoculated dentin specimens were rinsed in distilled water for 5 s. The specimens were divided into two groups: one group was kept moist, while the other was air-dried for 5 s. Resin containing 10 wt% RG was applied to the top surface of the dentin specimens with a sterile microbrush (Microbrush Fine, Shofu, Kyoto, Japan) and left for 30 s, followed by 10 s light-curing (Pencure 2000, Morita, Kyoto, Japan) with a light intensity of 2000 mW/cm^2^. The dentin specimens were rinsed with distilled water and manually divided into two pieces. The surface of the cross-sections was stained using the LIVE/DEAD^™^ Bac-Light^™^ Bacterial Viability Kit (L-7007, Thermo Fisher Scientific) according to the manufacturer’s instructions. Briefly, 2 µL of component A (SYTO 9 dye, 1.67 mM/propidium iodide, 1.67 mM solution in DMSO) and 2 µL of component B (SYTO 9 dye, 1.67 mM/propidium iodide, 18.3 mM solution in DMSO) were mixed in 1 mL of distilled water. One hundred microliters of the mixed solution was dropped on the specimens and incubated at 37 °C for 15 min in the dark. After gentle irrigation with distilled water, the specimens were visualized using confocal laser scanning microscopy (CLSM; LSM 700, Carl Zeiss, Oberkochen, Germany) with excitation at 488 and 555 nm and emission at 500 and 635 nm. Images were captured using ZEN Imaging Software (Carl Zeiss). A preliminary image was taken to set the acquisition parameters, which were kept constant for all images. Scans were performed in 12-bit mode at a resolution of 1024 × 1024 with the following settings: speed = 8; pinhole size = 34 µm; digital offset = 0; and master gain (Ch1, SYTO 9) = 493; master gain (Ch2, propidium iodide) = 706. Three images were obtained from each sample and analyzed using Imaris software (Bitplane, Zurich, Switzerland). Resin without RG served as the control, and the experiment was repeated three times.

### 4.10. Statistical Analysis

All statistical analyses were performed using SPSS Statistics version 25 (IBM Corp., Armonk, NY, USA). Homogeneity of variances was first confirmed. The solubility and ion release results were analyzed using analysis of variance (ANOVA) followed by Tukey’s honest significant difference (HSD) test, with a significance level of *p* < 0.05. Bacterial growth was analyzed using Student’s *t*-test, also with a significance level of *p* < 0.05.

## 5. Conclusions

The rapidly soluble Zn^2+^-releasing glass was successfully fabricated. The glass particles released high concentrations of Zn^2+^ quickly under wet conditions, regardless of pH, which is effective for eliminating oral bacterial species prevalent in deep caries. When incorporated into dental resin at a ratio of 10 wt%, the experimental resin demonstrated strong inhibitory effects on bacteria within the dentinal tubules. The novel phosphate-based glass developed in this study shows potential for use in cavity liners or pulp-capping materials that aid in cavity disinfection.

## Figures and Tables

**Figure 1 molecules-29-05098-f001:**
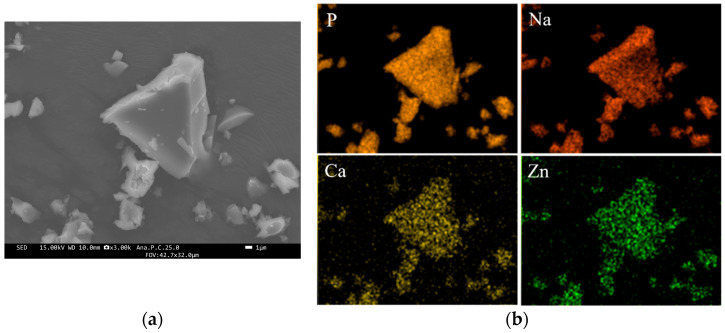
Field-emission scanning electron microscope (**a**) and elemental mapping (**b**) images of RG.

**Figure 2 molecules-29-05098-f002:**
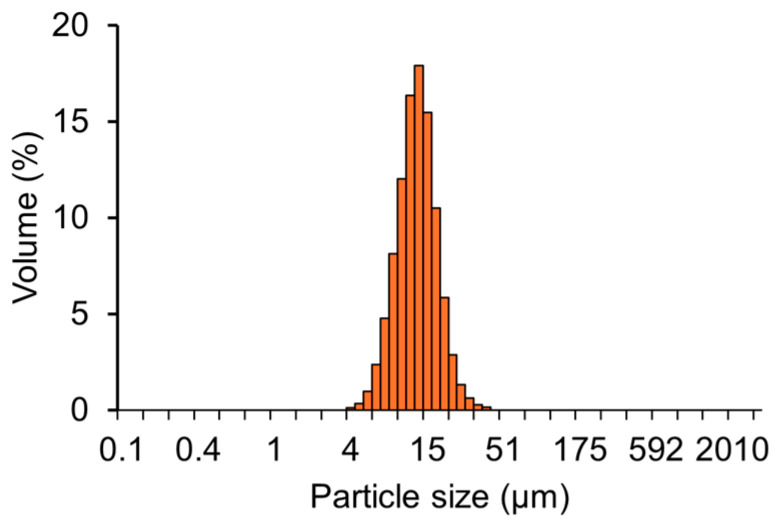
Particle size distribution of RG.

**Figure 3 molecules-29-05098-f003:**
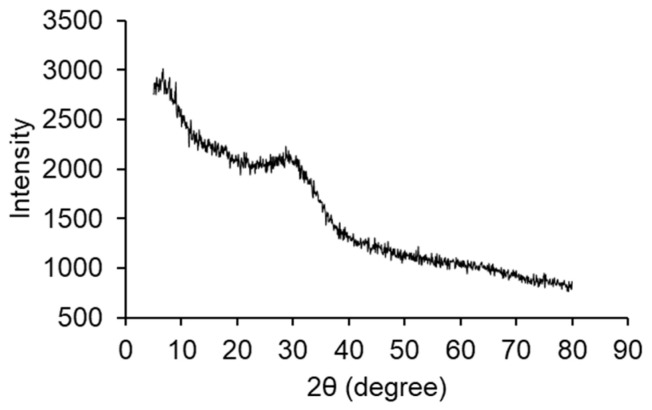
X-ray diffraction pattern of RG.

**Figure 4 molecules-29-05098-f004:**
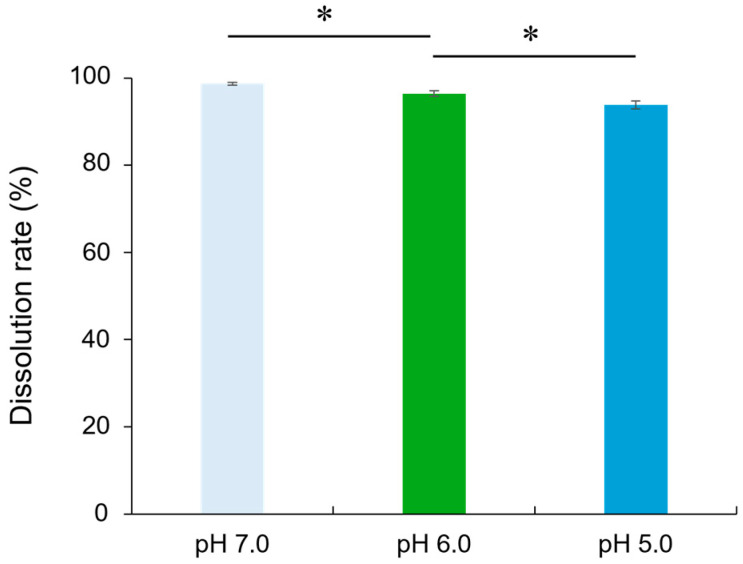
Solubility of RG in the pH-adjusted buffer solution (pH 7.0, 6.0, or 5.0). Bars represent the standard deviation of five replicates. * indicates significant differences (*p* < 0.05, ANOVA, Tukey’s HSD test).

**Figure 5 molecules-29-05098-f005:**
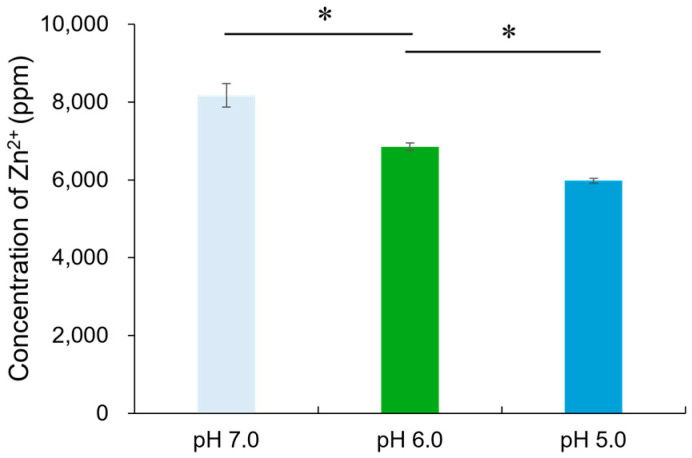
Release of Zn^2+^ from RG exposed to pH-adjusted BHI broth (pH 7.0, 6.0, or 5.0). Bars represent the standard deviation of five replicates. * indicates significant differences (*p* < 0.05, ANOVA, Tukey’s HSD test).

**Figure 6 molecules-29-05098-f006:**
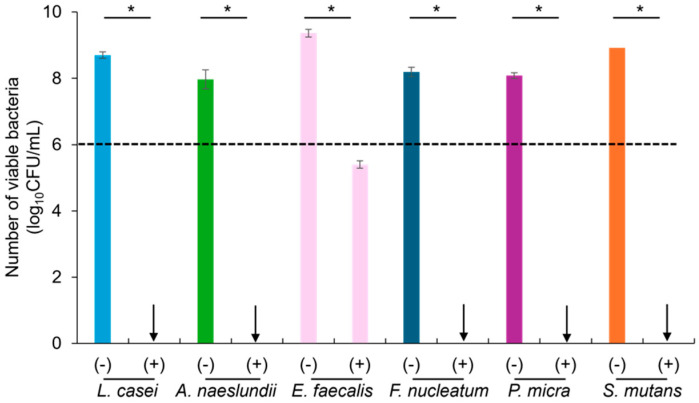
Number of viable bacteria after incubation in the absence (−) and presence (+) of RG. Arrows indicate 100% killing effect. Bars represent the standard deviations of five replicates. * indicates significant differences (*p* < 0.05, Student’s *t*-test). The dashed line indicates the initial amount of bacteria.

**Figure 7 molecules-29-05098-f007:**
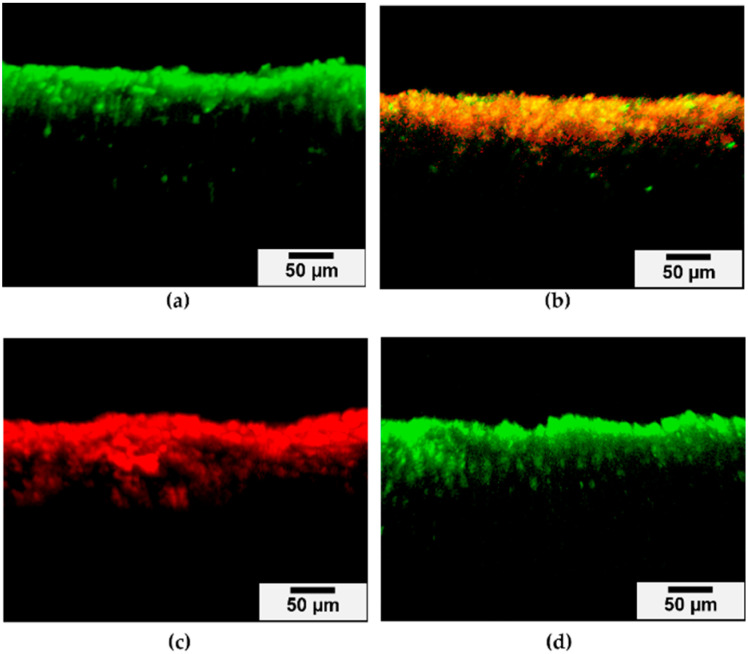
CLSM images of *L. casei*-infected dentin model (**a**), *L. casei*-infected dentin model treated by dental resin incorporating 10 wt% RG under the dry condition (**b**), *L. casei*-infected dentin model treated by dental resin incorporating 10 wt% RG under the wet condition (**c**), and *L. casei*-infected dentin model treated by control resin without RG (**d**) after LIVE/DEAD staining. Scale bar, 50 μm.

**Figure 8 molecules-29-05098-f008:**
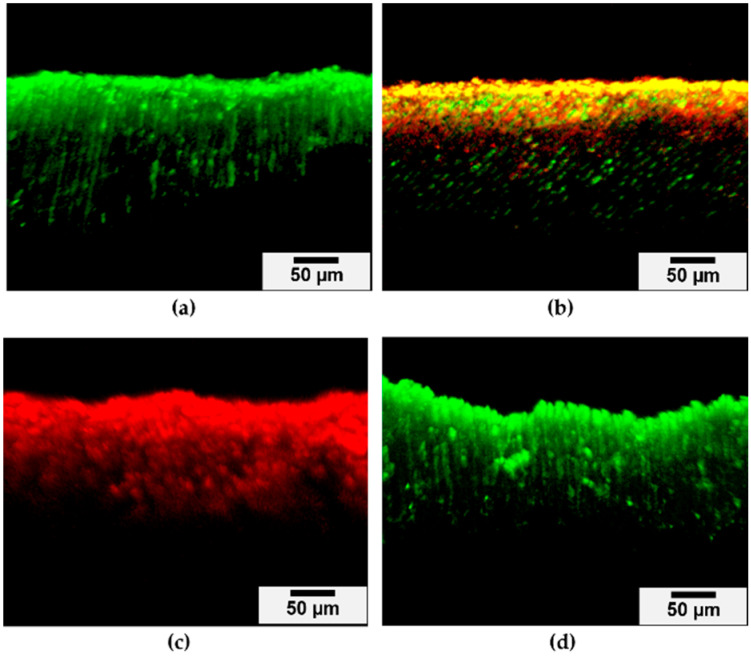
CLSM images of *S. mutans*-infected dentin model (**a**), *S. mutans*-infected dentin model treated by dental resin incorporating 10 wt% RG under the dry condition (**b**), *S. mutans*-infected dentin model treated by dental resin incorporating 10 wt% RG under the wet condition (**c**), and *S. mutans*-infected dentin model treated by control resin without RG (**d**) after LIVE/DEAD staining. Scale bar, 50 μm.

**Table 1 molecules-29-05098-t001:** Minimum inhibitory concentrations (MICs) and minimum bactericidal concentrations (MBCs) of Zn^2+^ against oral bacteria.

Species	MIC (ppm)	MBC (ppm)
*Lactobacillus casei* ATCC4646	500	2000
*Actinomyces naeslundii* ATCC19246	62.5	125
*Enterococcus faecalis* SS497	1000	4000
*Fusobacterium nucleatum* 1436	125	500
*Parvimonas micra* GIFU7745	125	250
* *Streptococcus mutans* NCTC10449	125	250

* Note: the MIC/MBC values for Streptococcus mutans NCTC10449 were from our previous study [19].

**Table 2 molecules-29-05098-t002:** Bacteria and culture conditions.

Species	Broth	Agar
*Lactobacillus casei* ATCC4646	Lactobacilli Inoculum Broth	Lactobacilli Inoculum Broth plates supplemented with 1.5% Bacto agar
*Actinomyces naeslundii* ATCC19246	Brain Heart Infusion Broth	Brain Heart Infusion agar
*Enterococcus faecalis* SS497	Brain Heart Infusion Broth	Brain Heart Infusion agar
*Fusobacterium nucleatum* 1436	Todd Hewitt Broth containing 0.1% L-cysteine	Todd Hewitt Broth agar containing 0.1% L-cysteine
*Parvimonas micra* GIFU7745	Brain Heart Infusion Broth	Brain Heart Infusion agar
*Streptococcus mutans* NCTC10449	Brain Heart Infusion Broth	Brain Heart Infusion agar

**Table 3 molecules-29-05098-t003:** Components of uncured resin.

Components	Function	wt%
Triethylene glycol dimethacrylate (TEGDMA)	Monomer	89.0
2-hydroxyethyl methacrylate (HEMA)	Monomer	10.0
Camphorquinone (CQ)	Initiator	0.3
Ethyl p-dimethylaminobenzoate (EPA)	Reducing agent	0.6
2,6-di-tert-butyl-p-benzoyl (BHT)	Inhibitor	0.1

## Data Availability

Data are contained within the article.

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
