# Peer review of "Fabrication of Rapidly Soluble Zn2+-Releasing Phosphate-Based Glass and Its Incorporation into Dental Resin"

_molecules, 2024, doi:10.3390/molecules29215098_

Round 1
Reviewer 1 Report
Comments and Suggestions for Authors
This paper examines an important aspect for the improvement of composites. It has been shown in studies conducted over 15 years It therefore makes sense to equip composites with the appropriate properties to inhibit or completely destroy these bacteria and to prevent the formation of marginal gaps through esterase activity.
The study was carefully planned and executed. The results are impressive and clearly show that the zinc oxide-enriched, rapidly soluble glasses are very well suited to completely destroy the examined bacteria, or in the case of E. faecalis, partially destroy them through their bactericidal properties.
The cited literature is adequate, the figures are easily recognizable and clearly arranged. The paper of this elaborate study seems suitable for publication.
Thank You.
Author Response
We would like to express our heartfelt thanks for your thoughtful and positive feedback on our work. We truly appreciate your recognition of the results and their potential implications for improving resin composites. Thank you.
Reviewer 2 Report
Comments and Suggestions for Authors
The aim of the paper is to fabricate a rapidly soluble Zn-ion-releasing phosphate-based glass (RG) especially designed for use as dental cavity liners and to evaluate its ion-releasing properties and antibacterial effects. Additionally, a dental resin incorporating RG was fabricated to serve as a cavity liner, and its effectiveness was investigated in vitro.
The topic of the research is up-to-date and corresponds to the journal area. The abstract is informative and the structure of the manuscript is well designed. Adequate research methods are used for characterization of the samples and evaluation of their properties. The conclusions are relevant to the results obtained. All figures, tables and references are cited in the text. There are not missing tables, figures and references.
The novelty of the study consists of the development and fabrication of a rapidly soluble Zn-ion-releasing phosphate-based glass (RG) especially designed for use as dental cavity liners with additional incorporating in a dental resin.
Advantages: this study is fully completed – the new materials is developed and its composition, structure and properties such as solubility, ion-release and antibacterial activity are investigated in-vitro.
Limitations: additional in-vivo tests are needed for approval of antibacterial properties of the new developed material and optimization of its composition.
Significance: This research has not only scientific value but is also of wide interest to dental clinicians because the novel phosphate based glass shows potential for use in cavity liners or pulp capping materials that aid in cavity disinfection.
Author Response
We would like to sincerely thank the reviewer for the positive and encouraging feedback on our study. We greatly appreciate your recognition of the novelty and significance of our research, as well as the thorough evaluation of our methods and results. Regarding the limitation noted, we fully agree that additional in vivo tests are necessary to further validate the antibacterial properties of the developed material and to optimize its composition. We plan to address this in future research to advance the clinical application of this material. Thank you.
Reviewer 3 Report
Comments and Suggestions for Authors
The article titled "Fabrication of rapidly soluble Zn-releasing phosphate-based glass and its incorporation into dental resin" (molecules-3237384) is of high quality and does not require additional experiments. The manuscript is written in a clear and accessible manner, and the employed methodology is sound. Therefore, I recommend it for publication. However, I have three suggestions for improvement:
-
I suggest the authors use the units recommended by IUPAC and the International System of Units (SI) instead of ppm. Providing information on how many milligrams of Zn are contained and released from 1g of RG would make the results clearer.
-
In the caption for Figure 6, the (+) and (-) symbols may have been mixed up. It currently states: ‘Number of viable bacteria after incubation in the presence of RG. (-): After incubation in the absence of RG. (+)’
-
It would be interesting to discuss whether phosphate-based glass without Zn could affect bacterial growth, as this was not investigated in the study.
Author Response
We would like to express our gratitude to you for the helpful critique and constructive suggestions regarding our manuscript. We have revised our manuscript according to the comments and suggestions raised by you.
- I suggest the authors use the units recommended by IUPAC and the International System of Units (SI) instead of ppm. Providing information on how many milligrams of Zn are contained and released from 1g of RG would make the results clearer.
Response: We understand the importance of using IUPAC and SI units for clarity. However, ppm is commonly used in materials science when discussing trace amounts of substances like metal ions. For example, ppm is frequently used to express the concentration of elements released from solid materials into liquids. For very low concentrations, ppm can be easier to conceptualize for readers. Since 1 ppm is equivalent to 1 mg/L (for dilute aqueous solutions), it directly indicates the mass of a substance in relation to a specific volume. This can be particularly useful for understanding low concentrations of ions without converting to moles. In addition, ppm is the standard in similar studies, so it’s reasonable to retain it to align with the conventional reporting in the field of dental materials, helping readers easily grasp our results in relation to existing data. That said, we appreciate the reviewer’s suggestion and will consider including data in future studies that indicates the milligrams of ions released from 1g of glass, as this may further broaden the accessibility of our findings for a wider audience.
- In the caption for Figure 6, the (+) and (-) symbols may have been mixed up. It currently states: ‘Number of viable bacteria after incubation in the presence of RG. (-): After incubation in the absence of RG. (+)’
Response: Thank you for pointing this out. We have corrected the caption for Figure 6 (Line 117).
- It would be interesting to discuss whether phosphate-based glass without Zn could affect bacterial growth, as this was not investigated in the study.
Response: Silicate-based bioactive glasses release silica, calcium, and sodium ions when the glasses dissolve in the aqueous environment [1]. This process typically results in a pH increase, making the solution more alkaline, which contributes to a certain degree of antibacterial effect. However, phosphate-based glasses show a slight pH decrease upon dissolution [2], and therefore, pH-related antibacterial effects are not expected. To achieve antibacterial properties, phosphate-based glasses need to be doped with antibacterial ions such as Zn [3]. For this reason, we did not evaluate the antibacterial properties of zinc-free phosphate-based glasses in this study. However, we agree that using zinc-free phosphate-based glass as a control could yield interesting results in future studies. Thank you for your suggestion, and we will consider this in future research.
[1] Begum S, Johnson WE, Worthington T, Martin RA. The influence of pH and fluid dynamics on the antibacterial efficacy of 45S5 Bioglass. Biomed Mate. 2016; 11: 015006.
[2] Abou Neel EA, Chrzanowski W, Pickup DM, O'Dell LA, Mordan NJ, Newport RJ, Smith ME, Knowles JC. Structure and properties of strontium-doped phosphate-based glasses. J R Soc Interface 2009; 6: 435-46.
[3] Liu L, Pushalkar S, Saxena D, LeGeros RZ, Zhang Y. Antibacterial property expressed by a novel calcium phosphate glass. J Biomed Mater Res B Appl Biomater 2014; 102: 423-429.